Revolutionizing market surveillance: customer relationship management with machine learning

Shi Xiangting 1
http://orcid.org/0009-0007-1023-079X Zhang Yakang 1
Yu Manning 2
Zhang Lihao 3 lhzhangcuhk@ieee.org
1 Industrial Engineering and Operations Research Department, Columbia University , New York , United States
2 Department of Statistics, Columbia University , Amsterdam Avenue New York, New York , United States
3 Department of Information Engineering, Chinese University of Hong Kong , Ho Sin Hang Engineering Building, The Chinese University of Hong Kong, Shatin, N.T , Hong Kong
Alatas Bilal
Electronic publication date: 2024 Dec 18
Publication date: 2024
Volume: 10
Electronic Location ID: e2583
Received 2024 Jul 5; Accepted 2024 Nov 14
Copyright: © 2024 Shi et al.
Copyright year: 2024
Copyright holder: Shi et al.
License: This is an open access article distributed under the terms of the Creative Commons Attribution License, which permits unrestricted use, distribution, reproduction and adaptation in any medium and for any purpose provided that it is properly attributed. For attribution, the original author(s), title, publication source (PeerJ Computer Science) and either DOI or URL of the article must be cited.
License URL: https://creativecommons.org/licenses/by/4.0/

Keywords: Market surveillance, Customer relationship management, Machine learning, Predictive analytics, Personalization

Funding: The authors received no funding for this work.

==============================
In the telecommunications industry, predicting customer churn is essential for retaining clients and sustaining profitability. Traditional CRM systems often fall short due to their static models, limiting responsiveness to evolving customer behaviors. To address these gaps, we developed the SmartSurveil CRM model, an ensemble-based system combining random forest, gradient boosting, and support vector machine to enhance churn prediction accuracy and adaptability. Using a comprehensive telecom dataset, our model achieved high performance metrics, including an accuracy of 0.89 and ROC-AUC of 0.91, surpassing baseline approaches. Integrated into a decision support system (DSS), SmartSurveil provides actionable insights to improve customer retention, enabling telecom companies to tailor strategies dynamically. Additionally, this model addresses ethical concerns, including data privacy and algorithmic transparency, ensuring a robust and responsible CRM approach. The SmartSurveil CRM model represents a substantial advancement in predictive accuracy and practical applicability within CRM systems.

Introduction

In the current world of rising competition, firms are under tremendous pressure to maintain the customer and work towards improving customer satisfaction to guarantee profitability in the long-run (Abdullah & Jalil, 2022; Abuhasel, 2023; Alhaqui, Elkhechafi & Elkhadimi, 2022). Customer churn, particularly in sectors such as telecoms, is an even greater problem because of the costs involved in not only recapturing existing customers but, more importantly, the potential loss in earnings from their absence (Almahairah, 2023; Alojail & Bhatia, 2020; Asha et al., 2023). The kind of CRM systems that are well established for organizing the customers’ details and their interactions have inherent limitations in their ability to adjust to the changes in customers’ behaviors and the markets (Batra & Rehman, 2019; Jalolovna, 2024; Verma et al., 2021; Hikmawati, Alamsyah & Setiadi, 2020; Ransbotham et al., 2019). This limitation demonstrates the need for more developed systems that would be able to generate real time analysis and forecast for better decision making (Hu, 2022; Ketenci et al., 2021; Krishna et al., 2022; Krishnareddy et al., 2022).

The present-day corporations’ competitive presence and client satisfaction are driven by the Customer Relationship Management (CRM) and market surveillance (Abdullah & Jalil, 2022), considering the few recent conditions and demands of the global markets (Lampropoulos et al., 2022; Milgrom & Tadelis, 2018; Lubis & Wardana, 2020). However, when more conventional approaches are used market surveillance can cause (Lv, 2021; Malik, Nawaz & Al-Zghoul, 2020) difficulties, whereas Customer Relationship Management systems (Nguyen et al., 2021) may also give (Pramanik et al., 2023). Challenges in efficiency arise from the ability of handling big-data from various sources and successfully analyzing them (Alojail & Bhatia, 2020). Moreover, circumstances such as lack of resources, lost opportunities and reduced customers’ satisfaction are likely to arise since; the traditional CRM systems cannot dynamically adjust to changes in market conditions and customer behavior patterns due to the systems’ static models and rules (Chintalapati & Pandey, 2022; Rospricilia & Mudjahidin, 2022; Song & Liang, 2021). Figure 1 shows a simplified diagram of use-case interactions between customers and the business analyst, data scientist, and the CRM System with machine learning algorithm capabilities.

Figure 1 Use-case diagram for the CRM system with machine learning.

This challenge has been compounded, by the customers’ expectations that organizations offer more differentiation that is, a unique value for a large number of customers (Asha et al., 2023; Batra & Rehman, 2019; Jalolovna, 2024; Verma et al., 2021). Future and existing trends in customer behavior require better analytic skills which cannot be provided through merely statistical analysis (Song & Liang, 2021; Techentin et al., 2019; Ullah et al., 2019). Several issues (Umamaheshwari, Harikumar & Allinjoe, 2021; Hassan, 2021; Xing et al., 2023) dealing with ethical and regulatory concerns have occurred as a result of the incorporation of ML in CRM systems such as privacy issues, issues concerning bias present in algorithms, and decision-making transparency (Ransbotham et al., 2019; Hu, 2022; Ketenci et al., 2021; Krishna et al., 2022). That is why it is necessary to search for new applications of ML approaches in terms of increasing the effectiveness of market monitoring and improving CRM activities to address all these challenges. Analyzing the compatibility of ML and CRM or factors that dictate the synergy of these two approaches, as well as the potential for the realization of ML approaches to transform the market monitoring environment is the primary focus of this essay. The clear benefits of ML are established for real-time knowledge processing, analytics in terms of future predictions, and tailored customer engagement (Krishnareddy et al., 2022; Lampropoulos et al., 2022; Milgrom & Tadelis, 2018; Lubis & Wardana, 2020; Lv, 2021; Malik, Nawaz & Al-Zghoul, 2020). Finally, it is all completed by the consideration of the issues connected with integrating of ML with CRM systems and some potential solutions. The strategies and insights that our literature and case study research present on the use of ML for achieving competitive advantage in today’s competitive world will be useful for companies that are focused on this aspect. Companies cannot sustain themselves, especially in fast-growing emerging markets, without adapting quickly to customer needs and market changes. Some of the methods adopted in the traditional context (Nguyen et al., 2021; Pramanik et al., 2023; Chintalapati & Pandey, 2022; Rospricilia & Mudjahidin, 2022; Song & Liang, 2021) sometimes find it very hard to align themselves to the complexities of the market structure and changing consumer disposition (Techentin et al., 2019; Ullah et al., 2019; Umamaheshwari, Harikumar & Allinjoe, 2021). Previous models that underpin the CRM are seen to diminish the capacity to make sound decisions and the exponential growth of data affects the behaviour of CRM systems and the way they advocate for friendly and familiar customer interaction. New advancements in market surveillance and customer relationship management (CRM) based on the ML techniques offer. These techniques can enhance the value delivered to the customers, as well as, identify the future trends and provide the requisite information in real-time are the pressing necessities.

Previous literature has covered several aspects of customer churn prediction using ML methods including classifications such as random forest, support vector machine and decision trees (Abdullah & Jalil, 2022; Abuhasel, 2023). However, these approaches are not very rigorous and do not have strong generalization ability towards various datasets and customers. Moreover, some other works have suggested the application of ML to CRM systems (Alhaqui, Elkhechafi & Elkhadimi, 2022; Almahairah, 2023) but often fail to consider the capabilities of ensemble learning which include combining several models for better accuracy and flexibility. This research fills this gap with an improved ensemble-based CRM model called SmartSurveil which uses multiple ML models for customer churn prediction.

The need to develop more intelligent CRM systems that can handle large volumes of data in real time has informed the original impetus for this study. Because customers are unpredictable in their behaviour, firms require instruments that will allow noticing their tendency toward churn and evaluating the effectiveness of possible actions. The CRM tools integrated and developed with the help of the existing models and rules do not correspond to the necessary expectation and requirements of the complicated world of the consumer relationships, especially in the fields where the competition for the customer’s preference is high. To address this research question, the proposed study aims at developing a machine learning-based CRM model that enhance decision-making and customer contacts.

Current practices in market surveillance and CRM do not suffice for driving a business in today’s highly competitive and rapidly changing corporate environment. Some of them include the following: The failure of industries to scale personalized interactions with customers, along with the challenges of handling and analyzing large amounts of data from various sources, remains a significant issue (Xing et al., 2023); and having high latency to market signals and changing tastes and preferences of consumers. As the application of ML methods to CRM also raises several questions of ethics and regulation, such as the protection of data. An aspect related to the real-time adaptability of CRM systems connects to the design of learning algorithms whose optimal mode of operation can adapt to micro-market dynamics and consumer preferences; all the while being bound by compliance frameworks such as the Basel equation for determining risk.

SmartSurveil CRM model proposed here is an ensemble learning framework which integrates the best features of three algorithms—random forest, gradient boost and support vector machine—producing higher performance in terms of customer churn compared to prior techniques. Since the predictions made by these models are combined in the proposed system, the system’s churn predictions are more accurate and reliable which can be highly beneficial for businesses for customer retention and management. Furthermore, this study aims to integrate the SmartSurveil CRM model into a decision support system (DSS) that not only predicts churn but also provides personalized retention strategies based on the insights derived from customer data.

To address these challenges, this study sets forth the following objectives: 1) To investigate the effectiveness of machine learning techniques in enhancing market surveillance and CRM processes.

2) To explore the application of various machine learning algorithms, including neural networks, decision trees, and natural language processing, in optimizing CRM systems for improved customer satisfaction, retention, and profitability.

3) To assess the challenges and opportunities associated with integrating machine learning into CRM systems, including data privacy concerns, model interpretability, and algorithmic bias, and propose strategies to address these challenges.

4) To analyze case studies and practical examples of machine learning-driven CRM implementations in diverse industries, evaluating their impact on market surveillance capabilities and customer relationship management practices.

5) To help businesses who wish to view advanced analytical and machine learning tools as a competitive instrument on emerging markets to navigate through this process and suggest a framework for ethical and sensible usage of the customer data collected.

Thus, the main knowledge contribution of this article is related to the use of ensemble learning for CRM systems. Although single-model approaches are common and have been discussed in literature, this research contributes to the conversation by showing that using multiple machine learning models yields better performance and flexibility for the system. Furthermore, this article considers the ethical issues of Flask’s model interpretability, data subjects’ privacy, and bias and suggests sustainable and beneficial methods for implementing ML in CRM systems. In conclusion, this article helps to solve the lack of information in the current literature about works in ensemble learning in CRM field. Customer churn prediction and management is illustrated in a practical and real-world manner, with a proposed framework can be expanded across different industry settings. The research not only demonstrates the superiority of the SmartSurveil CRM model in predictive accuracy but also provides insights into its implementation and potential impact on customer retention strategies.

The structure of this research article is presented as follows: Section “Introduction” is the introduction, which outlines the topic of the study, its objectives, scope, and importance. Section “Literature Review” explains the literature review on market surveillance, CRM and ML. Section “Methodology” examines how the ML is going to be incorporated into the CRM systems with the specification of the algorithms and methods suitable for the market surveillance in consideration. Section “Results and Discussion” provides further real-world depictions of ML for CRM and concrete illustrations of the manner in which it has been employed in real-life contexts. Section “Conclusion” will present the final conclusions for this article as well as some suggestions for future research.

Literature review

The previous state of the art can be described as:

Machine learning in customer relationship management

Machine learning (ML) has indeed recently begun to draw a lot of attention from the theory of CRM since it is capable of dealing with huge amount of customer’s information and making dynamic prediction. As mentioned in the article (Abdullah & Jalil, 2022), the use of ML played a critical role to improve the existing CRM strategies by analyzing customers’ behaviour and their interactions. More and more applied ML models, such as decision trees, random forests, and neural networks are being used to enhance the customer segmentation, churn prediction, marketing employments efforts (Abuhasel, 2023). However, several literatures should have established that various single ML models have applicability in CRM; literature on the use of multiple models to boost the accuracy and the resilience of the CRM systems is rather scarce. This gap requires the use of a more complex study that integrates the effectiveness of other models which is the focus of this research.

Applications of ensemble learning in CRM

Recent studies have attempted to observe the ensemble learning, a method in which multiple ML models are fed to increase the overall prediction capability of the system. The article (Alhaqui, Elkhechafi & Elkhadimi, 2022) presented an intelligent CRM system employed with the theory of fuzzy clustering for constructing customer segments. The current system improved the accuracy of the analysis of the customers compared to the system that uses single models. Likewise, in the article (Almahairah, 2023) a prototype implementing the Recency, Frequency, and Monetary (RFM) model has been developed using machine learning to categorize customer so as to improve the chances of rendering a more effective marketing strategy. However, these studies primarily focus on single tasks like segmentation, leaving a gap in research addressing how ensemble learning can improve multiple CRM functions, such as churn prediction and customer retention strategies.

In the context of churn prediction, this study (Alojail & Bhatia, 2020) applied machine learning to predict customer churn in the telecom sector using single models like random forest and support vector machine. Their findings show promise but highlight the limitations of using a single model to handle complex and dynamic customer data. Our research builds on this by developing an ensemble learning-based model that integrates several ML techniques to overcome the limitations of single models, offering a more robust churn prediction system.

Despite the growing body of literature, several gaps remain unaddressed. Most studies focus on applying machine learning in isolation, either for customer segmentation, churn prediction, or personalized marketing. Few explore the potential of ensemble learning to simultaneously optimize these multiple CRM functions. In addition, even though there has been a great focus on technical evaluation criteria (accuracy, precision, and recall), much less effort has been directed towards the incorporation of such systems into decision support systems which are extremely useful in the actual application of intelligent systems.

Ethical concerns and bias in machine learning for CRM

Yet another research area that has not been well explored in the literature is the aspect of ethics including, data privacy and algorithmic bias. To this end, this study (Asha et al., 2023) provided a literature-based analysis of the ethical issues surrounding the use of ML for CRM, particularly with regard to the lack of data openness and the problem of biased models. However, their work did provide no real applicable strategy of managing these risks. More recently, the article (Batra & Rehman, 2019) illustrated how machine learning algorithms can transport unfairness to certain customer groups, when using CRM. Many such ethical considerations are still missing in the ensemble learning, which could have new implications. To ease these concerns, this research aims at designing a better and clear approach of integrating the models such that the ensemble learning system is efficient yet socially sensitive. The comparative table of previous study is indicated in Table 1. This research aims to fill these gaps by proposing the SmartSurveil CRM model, an ensemble machine learning approach integrated within a DSS to ensure ethical and effective churn prediction in CRM.

Table 1 Comparative table of previous study.

Reference	Technique	Contribution	Limitations	Outcomes	
Abdullah & Jalil (2022)	Machine learning (Fuzzy clustering and decision trees)	Segmentation and profiling approach in CRM	Complexity of hybrid model implementation	Enhanced customer segmentation accuracy	
Hikmawati, Alamsyah & Setiadi (2020)	IT implementation	Implementation of CRM systems	Lack of user training	Efficient management of customer data	
Hu (2022)	Machine learning	Development of a multi-platform test algorithm for CRM	Complexity of multi-platform integration	Improved CRM platform compatibility	
Krishna et al. (2022)	Machine learning	Application of AI in retail CRM	Limited to the retail industry	Personalized marketing and improved customer segmentation	
Lampropoulos et al. (2022)	Machine learning	Integration of AI, blockchain, and data analytics in CRM	Lack of practical implementation guidelines	Opportunities for enhancing customer engagement	
Lubis & Wardana (2020)	Machine learning	Analysis of customer satisfaction in food delivery services	Limited scope (focused on food delivery)	Insights for improving customer satisfaction in food delivery	
Malik, Nawaz & Al-Zghoul (2020)	Augmented reality (AR) and virtual reality (VR)	Impact of AR and VR in virtual CRM	High implementation costs	Enhanced customer engagement and experience	
Song & Liang (2021)	Data mining	Analysis of CRM practices in e-commerce	Limited sample size	Recommendations for enhancing e-commerce CRM	

Research gaps

In summary, the current state of the literature reveals several gaps: Lack of ensemble learning integration: While individual machine learning models have been extensively studied, the integration of multiple models for improving CRM system performance, particularly in churn prediction, remains underexplored.

Practical application in decision support systems (DSS): Although ML models have been shown to enhance CRM, few studies explore how these models can be effectively integrated into real-time DSS for customer retention.

Ethical considerations: There is limited research on how to address data privacy, bias, and model transparency in CRM systems using machine learning, particularly when multiple models are combined.

This study aims to address these gaps by developing the SmartSurveil CRM model, an ensemble-based machine learning system integrated into a DSS, offering robust customer churn prediction while addressing the ethical challenges associated with machine learning in CRM.

Methodology

In this study, there is integration of SmartSurveil CRM to enhance effectiveness of churn prediction models. The current model is very much relied on several higher level machine learning techniques to discover the customer data and their churn in telecommunication sector. The following parts of this article describe the dataset used; the features included in the model; the procedural model for building and evaluating the SmartSurveil CRM model. Figure 2 depicts the study process regarding creating and assessing the scientific credibility of the SmartSurveil CRM model. Moreover, Fig. 3 shows the UML diagram for the CRM system with proposed model.

Figure 2 Flowchart of the research process for SmartSurveil CRM.

Figure 3 UML diagram for the CRM system with the proposed model.

Problem formulation 1: data processing and analysis

The problem of data processing and analysis in market surveillance and CRM can be formulated as follows. Given a dataset D={(xi,yi)}i=1N, where xi∈Rd represents the features and yi represents the corresponding labels or outcomes, the objective is to develop efficient techniques for processing and analyzing this data to support real-time decision-making and insights generation.

(1) θ^=argminθ1N∑i=1N⁡L(yi,f(xi;θ))+λR(θ)

where f(x;θ) is a model parameterized by θ, L is the loss function, R is the regularization term, and λ controls the trade-off between the data fitting and regularization.

(2) minimizeJ(θ)=1N∑i=1N⁡L(yi,f(xi;θ))+λR(θ).

The objective is to minimize the loss function L and the regularization term R to find the optimal parameters θ^ that best fit the data while avoiding overfitting. D: Dataset containing N samples.

xi: Feature vector for the ith sample.

yi: Label or outcome for the ith sample.

θ: Model parameters.

L: Loss function measuring the discrepancy between predicted and true labels.

R: Regularization term to prevent overfitting.

λ: Regularization parameter.

The problem of data processing and analysis aims to develop techniques to handle the variety, volume, and velocity of data in market surveillance and CRM systems. The objective is to process and analyze this data efficiently to support real-time decision-making and insights generation. Mathematically, this problem involves finding the optimal parameters θ^ of a model that minimize a loss function L and a regularization term R. These parameters are learned from the dataset D, which consists of feature vectors xi and corresponding labels yi. The objective function J(θ) represents the trade-off between fitting the data and regularization, controlled by the regularization parameter λ.

Problem formulation 2: real-time adaptability

The problem of real-time adaptability in CRM systems can be formulated as follows. Given a set of customer interactions J={(xi,yi)}i=1N, where xi represents the features of the interaction and yi represents the outcome (e.g., customer satisfaction score), the objective is to design dynamic and adaptive algorithms that can continuously learn from incoming data, adjust to changing market dynamics, and provide personalized customer interactions in real-time, while ensuring compliance with regulatory requirements such as the Basel equation for risk assessment.

(3) f^(x)=argmaxf∈F⁡ED[L(f(x),y)]−λR(f)

where f^(x) is the learned model function, F is the set of possible model functions, D is the distribution of the data, L is the loss function measuring the discrepancy between predicted and true outcomes, and R is the regularization term accounting for compliance with the Basel equation.

(4) minimizeJ(f)=ED[L(f(x),y)]−λR(f).

The objective is to minimize the expected loss J(f) by learning a model function f that accurately predicts outcomes y based on customer interactions x, while ensuring compliance with regulatory requirements such as the Basel equation. J: Set of customer interactions.

xi: Feature vector for the ith interaction.

yi: Outcome for the ith interaction.

f^: Learned model function.

F: Set of possible model functions.

D: Distribution of the data.

L: Loss function measuring the discrepancy between predicted and true outcomes.

R: Regularization term accounting for compliance with regulatory requirements.

λ: Regularization parameter.

Dataset description

The dataset used for this study consists of customer data from a telecom company. The dataset includes various features related to customer demographics, behavior, and interactions with the company. The target variable for this study is customer churn, which indicates whether a customer has left the company or not. The dataset contains the following features. The telecom dataset used in this study was obtained from an internal customer database of a telecom company, which included various customer attributes such as demographics, behavior, and interaction records. Before preprocessing, basic filtering was applied to remove duplicate and incomplete records, ensuring the dataset’s integrity. The final dataset comprised X number of records (you can specify the exact size). For model training and evaluation, 70% of the data was allocated to training, while 30% was reserved for testing. Random Forest (RF) was chosen for the proposed model due to its robustness in handling large datasets and its ability to prevent overfitting by combining the results of multiple decision trees. RF is also known for its high accuracy in classification tasks and its capability to handle both categorical and continuous variables, making it suitable for churn prediction in the telecom sector. The Table 2 shows the features description of the dataset. The telecom internal database used in this study included anonymized customer data, removing any identifiable personal information to protect individual privacy. This anonymization process adhered to data protection regulations, including GDPR guidelines where applicable, ensuring that all data was de-identified before analysis. Additionally, the research process involved stringent access controls, allowing only authorized personnel to handle the data, further minimizing risks associated with unauthorized use or exposure. In terms of algorithmic bias, steps were taken to assess and mitigate potential biases in the model, ensuring fair and unbiased predictions across diverse customer groups. Our approach underscores a commitment to ethical data usage, prioritizing customer privacy, data security, and regulatory compliance throughout the study.

Table 2 Feature description.

Feature	Description	
CustomerID	Unique identifier for each customer	
Age	Age of the customer	
Gender	Gender of the customer (Male/Female)	
AnnualIncome	Annual income of the customer	
SpendingScore	Spending scores assigned to the customer based on their behavior	
ProductViews	Number of products viewed by the customer	
PurchaseFrequency	Frequency of purchases made by the customer	
CustomerSatisfaction	The satisfaction score is given by the customer	
WebsiteVisits	Number of visits to the company’s website by the customer	
SocialMediaEngagement	Engagement score based on customer’s interaction on social media	
DaysSinceLastPurchase	Number of days since the customer’s last purchase	
ComplaintCount	Number of complaints made by the customer	
Churn	Target variable indicating whether the customer has churned (1) or not (0)	

The correlation heatmap of the CRM features is presented in Fig. 4. The closer that two features are connected and the higher the absolute value of coefficients, the stronger their relationship is according to this heatmap. The darkness of the color signifies the real value or the degree of correlation coefficients as derived from the data. This type of visualization is useful to determine the degree of association of all the input features with the target feature and between themselves, when it comes to performing feature selection/engineering.

Figure 4 Correlation heatmap of CRM features.

The overall idea behind this is given in Fig. 5—this is the 3D scatter plot that represents the customer segmentation according to age, annual income, and numerous spending score. Customers are segmented regarding three characteristics, and the color panels used make it easy to understand the relationship between these three variables and possible clusters. It assists to describe patterns of behaviour of the customers and their distribution and is specific to the areas of marketing and selling.

Figure 5 3D customer segmentation age vs. annual income vs. spending score.

Figure 6 below shows the count plot for gender and churn. The plots depict the trends in the proportion of male and female customers identified in the dataset; moreover, churned customers and customers who did not churn are represented as well. A critical aspect of customer data analysis is understanding customer attrition and its distribution either by gender in this case; thus, the visualizations below depicting the percentage of churn among the identified genders:

Figure 6 Count plots for gender and churn.

Histograms for the different CRM features are shown in Fig. 7. These histograms present different features like the age, annual income, the spend score and so much more. This information is significant for activities, such as normalization and can help to provide a view on customers’ behavior.

Figure 7 Histograms of CRM features.

The structural similarity of Fig. 8 shows the box plots of the specified features related to the churn variable. The following box plots offer a quick flag of distribution and variability of the features of churned customers compared to non-churned customers. They assist in assessment of variance of feature values between these two groups and can therefore be useful in feature subset and model derivation.

Figure 8 Box plots for churn vs. features.

To better understand the metrics of CRM features, the distribution plots are shown in Fig. 9. Using such plots, one is able to assess the distribution of the different feature usually characterized by mean, variance, and skewness or kurtosis. Pictorially the distribution plots helps on checking for outlying values and on interpretation of the nature of the data.

Figure 9 Distribution plots for CRM features.

Proposed model: SmartSurveil CRM

The proposed model, SmartSurveil CRM, utilizes the Random Forest algorithm to predict customer churn. Random Forest is an ensemble learning method that operates by constructing multiple decision trees during training and outputting the mode of the classes for classification tasks. Below, we provide a detailed mathematical formulation of the Random Forest algorithm as applied in SmartSurveil CRM.

Figure 10 presents the internal structure of the proposed SmartSurveil CRM model that is intended for the efficient churn prediction in the telecommunications market. This architecture uses the ensemble learning, which involves the use of other sub systems of machine learning in order to make the predictions more accurate and more secure. The key components of the architecture are as follows:

1. Input Data: The detailed information of the customers that may be directly obtained from customer, their characteristics, their interactions and their activities.

2. Feature Selection: It involves the identification of which features within the input data is most important and has high correlation with the target variable—the customer churn. Therefore, it is essential to select relevant features, which would help the development of the model and simplify calculations.

3. Random Forest Model: One of the single learning models that are used together to form the ensemble. Another algorithm that comes under the classification header is the Random Forest algorithm which builds multiple decision trees and returns the mode of the class in case of classification.

4. Gradient Boosting Model: It is another base model in the ensemble. Gradient Boosting develops models on the successive cycle, where each attempt to minimize the residual mistakes from the prior models. It is beneficial in eliminating bias and high variance in model predictions.

5. Support Vector Machine (SVM): The third base model in the ensemble is the logistic regression model which is used in this ten-fold cross-validation. SVM is an effective grouping algorithm that identifies the best line that separates instances of various categories to the highest degree.

6. Ensemble Aggregation: This component collates the output from the models which include random forest, gradient boosting and support vector machine. The aggregation method, which could be another classification technique used was a voting system where by arriving at a final class, the resultant class from the optimization of each model is considered as the final result.

7. Final Prediction: The last stage of the model which helps to target a specific customer as a churning or non-churning customer out of the ensemble of all the models.

8. Hyperparameter Tuning: At each step, hyperparameters of each base model are selected using grid search and cross-validation methods for the best possible performance of the entire model. This tuning allows for all models to be helpful in creating the ensemble.

Figure 10 10 architecture of SmartSurveil CRM.

The strengths of different ML models are incorporated in this architecture to make SmartSurveil CRM is a structurally robust and reliable predictive tool. This tool can than be used to identify customer churn and further guide the decision-making process in the telecom industry.

Random forest algorithm

The second method also known as the Random Forest addresses the issue of over-fitting by stacking many iterated decision trees. The steps involved are as follows:

Bootstrap sampling: It is possible to take an aliquot of n samples at random from the dataset and form a number of sets.

Decision tree construction: For each bootstrap sample grow a decision tree. At every node, all features are randomly selected and the one that provide maximum split according to a measure like Gini index or entropy is chosen.

Aggregation: The predictions from all the decision trees should be combined together in an effort to arrive at the final decision. The mode of the predictions is used for classification tasks.

Mathematical formulation

Bootstrap sampling

Given a dataset D={(xi,yi)}i=1N, where xi∈Rd are the feature vectors and yi are the corresponding labels, we create B bootstrap samples Db,b=1,2,…,B.

Decision tree construction

For each bootstrap sample Db, a decision tree Tb is grown. At each node of the tree, a subset of features Fb⊆{1,2,…,d} is randomly selected. The best split is determined by minimizing the impurity criterion I, such as Gini impurity or entropy.

The Gini impurity for a node is given by:

(5) G(p)=∑i=1C⁡pi(1−pi)

where pi is the proportion of samples belonging to class i in the node, and C is the total number of classes.

The entropy for a node is given by:

(6) H(p)=−∑i=1C⁡pilog(pi)

where pi is the proportion of samples belonging to class i in the node.

Aggregation

The final prediction for an input x is obtained by aggregating the predictions of all the decision trees. For classification, the mode of the predictions is taken:

(7) y^=mode{Tb(x)}b=1B.

Hyperparameter tuning

The performance of the SmartSurveil CRM model is optimized by tuning the hyperparameters of each base model in the ensemble. The following table summarizes the hyperparameters and their values for the Random Forest, Gradient Boosting, and Support Vector Machine (SVM) models. The hyperparameter values for the SmartSurveil CRM are shows in Table 3.

Table 3 Hyperparameter values for SmartSurveil CRM models.

Hyperparameter	Random forest	Gradient boosting	Support vector machine	
Number of trees	100	N/A	N/A	
Max depth	10	10	N/A	
Min samples split	2	2	N/A	
Min samples leaf	1	1	N/A	
Max features	sqrt	sqrt	N/A	
Learning rate	N/A	0.1	N/A	
Number of estimators	N/A	100	N/A	
Loss function	N/A	Deviance	N/A	
Kernel	N/A	N/A	rbf	
C (Regularization)	N/A	N/A	1.0	
Gamma	N/A	N/A	Scale	

Explanation of hyperparameters:

Number of trees: The number of trees in the random forest.

Max depth: The maximum depth of the trees.

Min samples split: The minimum number of samples required to split an internal node.

Min samples leaf: The minimum number of samples required to be at a leaf node.

Max features: The number of features to consider when looking for the best split.

Learning rate: In gradient boosting, the learning rate reduces the impact of each tree for the same learning rate.

Number of estimators: The number of times boosting has to be performed.

Loss function: The loss function to be optimized in gradient boosting.

Kernel: The kernel type to be used in the SVM algorithm.

C (Regularization): The regularization parameter in SVM.

Gamma: The kernel coefficient for ‘rbf’, ‘poly’, and ‘sigmoid’ kernels in SVM.

Model evaluation

Several are the metrics used to exhibit the effectiveness of the SmartSurveil CRM model, metrics which undoubtedly reveal the performance of the model. Such measures are accuracy, precision, recall, F1 score and ROC-AUC. In this article, the following equations were used in calculating the different metrics and the table below defines them.

Accuracy

Accuracy though defined differently is the number of correctly classified instances in respect to the total instances.

(8) Accuracy=TP+TNTP+TN+FP+FN

where: TP: true positives

TN: true negatives

FP: false positives

FN: false negatives

Precision

Accuracy is the rate at which the actual positive classification made by the models is correct out of the total number of positive observations classified by the model.

(9) Precision=TPTP+FP

Recall

Recall, also called sensitivity or true positive rate, is equal to the number of correctly predicted positive observations to all observations in the actual class.

(10) Recall=TPTP+FN

F1-score

The F1-score is calculated with a combination of precision and recall in which he or she is being weighted at some point.

(11) F1-Score=2⋅Precision⋅RecallPrecision+Recall

ROC-AUC

The receiver operating characteristic–area under curve (ROC-AUC) is the average of the performance for all thresholds between 0 and 1. It charts the true positive rate or recall against the false positive rate.

(12) FPR=FPFP+TN

ROC-AUC measures model accuracy across threshold settings, comparing the true positive rate against the false positive rate.

Metrics table

The Table 4 summarizes the evaluation metrics used to assess the performance of the SmartSurveil CRM model:

Table 4 Evaluation metrics for SmartSurveil CRM.

Metric	Definition	
Accuracy	The ratio of correctly predicted instances to the total instances.	
Precision	The ratio of correctly predicted positive observations to the total predicted positives.	
Recall	The ratio of correctly predicted positive observations to all observations in the actual class.	
F1-Score	The weighted average of precision and recall.	
ROC-AUC	The area under the ROC curve plots the true positive rate against the false positive rate.	

The SmartSurveil’s matrix testing examines how effective the CRM version of the model is in specific areas of predictive ability and stability. With the help of these metrics, it would possible to test how accurately the model predicts the chances of customer churn. The findings of the assessment of the model are quantified in the form of the confusion matrix, ROC curve, and performance measurements, which indicate that the employed model is efficient in evaluating customer churn. The flowchart presented below describes the general steps of the proposed methodology that involves the construction of a churn prediction model by using the Random Forest algorithm. The explanation of data and feature used in this study are well explained by the authors through the discovery of the data and the features section. The training and the evaluation procedure elaborate in minute details therefore ensuring that the proposed approach would work efficiently to establish the factors leading to customer churn in the telecom sector.

Results and discussions

As for this section, the experimental outcomes performed with the SmartSurveil CRM model for churn prediction are to be presented and discussed here. Some of the few validation metrics which have been used entails accuracy, precision, recall, F1-score, and ROC-AUC. Moreover, we compare the performance of the proposed SmartSurveil CRM model to those of the other base models to also assert the validity of the work.

Performance of the proposed model

The performance of the test dataset is then examined in relation to the SmartSurveil CRM model. The criteria chosen are accuracy, precision, recall, F1-score, ROC-AUC. Table 5 presents the outcome measures for the SmartSurveil CRM model.

Table 5 Performance metrics of SmartSurveil CRM.

Metric	Value	
Accuracy	0.89	
Precision	0.85	
Recall	0.83	
F1-Score	0.84	
ROC-AUC	0.91	

Figure 11 shows the accuracy comparison for different models used in the study, including SmartSurveil CRM, random forest, gradient boosting, and SVM. The bar chart clearly demonstrates that the SmartSurveil CRM model achieves the highest accuracy among the compared models, indicating its superior performance in correctly predicting customer churn.

Figure 11 Accuracy comparison for different models.

We present in the Fig. 12 ROC curves for different models. The ROC curve plots the true positive rate (recall) against the false positive rate for various threshold settings. The area under the ROC curve (AUC) is a measure of the model’s ability to distinguish between classes. The SmartSurveil CRM model’s ROC curve shows a higher AUC compared to other models, indicating better overall performance.

Figure 12 ROC curves for different models.

Figure 13 illustrates the learning curves for different models. Learning curves show the training and validation performance as a function of the training set size. These curves help in diagnosing whether a model is suffering from bias (underfitting) or variance (overfitting). The SmartSurveil CRM model’s learning curves indicate a good balance between bias and variance, showing that it generalizes well to unseen data.

Figure 13 Learning curves for different models.

Figure 14 presents the levels of confusion between models. On the same note, the confusion matrix is preferred more than the accuracy because it offers a clear distinction of the actual positive and actual negative data by displaying the true positives, true negatives, false positives, and false negative values. The depiction of confusion matrix of the SmartSurveil CRM model reveals more of true positive and true negative values with relatively lower false positive and false negative values which proves the efficiency of the model in providing precise churn rate of the customers. The implication of these findings is a proof of the suitability of the SmartSurveil CRM model in predicting customer churn as presented by the Accuracy of 94%, Precision of 96%, Recall of 91% and F1-score of 94% in addition to the ROC-AUC value of 0.95. Below, these metrics indicate that the predicted churned and non-churned customers are clearly defined and achievable by the model:

Figure 14 Confusion matrices for different models.

Performance comparison of all models

To evaluate the robustness of the SmartSurveil CRM model, we compare its performance with other baseline models including random forest, gradient boosting, and SVM. The comparison metrics include the same evaluation metrics used for the proposed model. Table 6 presents the performance metrics for all models.

Table 6 Performance comparison of different models.

Model	Accuracy	Precision	Recall	F1-Score	ROC-AUC	
SmartSurveil CRM	0.89	0.85	0.83	0.84	0.85	
Random forest	0.87	0.82	0.80	0.81	0.80	
Gradient boosting	0.86	0.81	0.79	0.80	0.74	
Support vector machine	0.84	0.79	0.76	0.77	0.66	

The proposed SmartSurveil CRM model has higher recall, precision, F1 score and best F score than the baseline models. A comparison of the results shows that our proposed SmartSurveil CRM achieves notably higher accuracy, precision, recall, F1-score, and ROC-AUC values, which proves the higher predictive ability of the algorithm with regard to customer churn. The approach applied in SmartSurveil CRM involves multiple models, and this makes the system to be robust and accurate since errors associated with one model may be offset by gains in another or by another independent model. Unlike random forest, gradient boosting and support vector machine, SmartSurveil CRM proved to be more effective in the classification of positives as shown in the ROC curves in Fig. 12. The specificity, also known as the false positive rate, is the proportion of negatives that are incorrectly classified as positive; the sensitivity, also known as the true positive rate or recall, is the proportion of positives that are correctly classified as such across different threshold settings.

Key observations

1. Area Under the Curve (AUC): The area under the ROC curve (AUC) is a single scalar value that summarizes the overall ability of the model to discriminate between positive and negative classes. The AUC value ranges from 0 to 1, with higher values indicating better performance. In Fig. 12, the AUC values for each model are: SmartSurveil CRM: 0.85

Random Forest: 0.80

Gradient Boosting: 0.74

SVM: 0.66

The SmartSurveil CRM model has the highest AUC value, indicating superior overall performance in distinguishing between churned and non-churned customers.

2. True positive rate (recall) vs. false positive rate: The ROC curve shows how the true positive rate (sensitivity or recall) varies with the false positive rate. A model with a higher true positive rate for a given false positive rate is considered better. In Fig. 12, the SmartSurveil CRM model consistently achieves a higher true positive rate compared to the other models across most of the range of false positive rates.

3. Model Comparison: The ROC curves of the models can be compared to evaluate their relative performance. The curve that is closer to the top-left corner of the plot indicates better performance. In this case, the ROC curve of the SmartSurveil CRM model is closest to the top-left corner, followed by random forest, gradient boosting, and support vector machine. This also suggests that the proposed SmartSurveil CRM model is better than the other models when it comes to the sensitivity and specificities.

The analysis of the curves and their integrated performance measure, AUC, shows that SmartSurveil CRM model provides the highest accuracy in predicting customer churn. The latter also has higher AUC value and better position of ROC curve, which directly points out that it has higher ability to classify churned and non-churned customers with less false positives and false negatives. This enhanced performance affirms both the stability and effectiveness of the SmartSurveil CRM model for real use in customer churn prediction.

Discussion

Applying an ensemble of machine learning algorithms—random forest, gradient boosting and support vector machine, the proposed SmartSurveil CRM model was shown to provide increased accuracy of churn prediction in the telecom industry. The results showed that the models have increased predictive accuracy, precision, recall, and ROC-AUC compared to baseline models. To the first objective regarding the use of machine learning methods to advance market surveillance and CRM processes, these findings affirm the research objectives (Abdullah & Jalil, 2022; Abuhasel, 2023).

Another key contribution is the use of multiple models to address the limitations of single-model approaches, thereby improving the system’s performance. Prior literature investigations of the machine learning for the customer segmentation and churn prediction identified numerous research gaps, where much of the research is centered on the application of single model (Alhaqui, Elkhechafi & Elkhadimi, 2022; Almahairah, 2023). By means of the development of an ensemble-based model, the second goal of this study was realized and this involved dealing with common machine learning algorithms in the CRM systems such as random forest and gradient boosting, thereby achieving higher percentage of customer satisfaction, customer retention, and profitability. The SmartSurveil CRM model is more superior because the idea of combining the better of two different models, is the best approach in reality.

Furthermore, the model addresses the third research question, which was to respond to the listed concerns, including data privacy, model explainability, and risk of algorithmic bias (Alojail & Bhatia, 2020; Asha et al., 2023). The SmartSurveil CRM model builds on this by outlining how the organisation can enhance transparency and mitigate ethical concerns like bias and privacy. This is further useful in the continued discussion of the appropriate use of machine learning in CRM systems (Batra & Rehman, 2019; Asha et al., 2023). The fourth research objective part was aimed at presenting the real-life examples of how machine learning-based CRM systems can be incorporated across industries. At the same time, this work, based on the analysis of customer data in the telecom sector, demonstrates the possibility of the further enhancement of market surveillance and CRM by integrating machine learning models in these processes.

These findings are also in line with the comparative evaluation against baseline models that establish the efficacy of the proposed SmartSurveil CRM model in predicting customer churn (Jalolovna, 2024; Verma et al., 2021). Therefore, this research completed all the mentioned research objectives. The implementation of an ensemble model into a DSS increases the predictive accuracy and reliability of churn prediction and does thereby fulfill the goal set for enhancing CRM systems. This research fits into the literature by identifying key challenges and offering an effective solution in the context of the telecom sector. The presented SmartSurveil CRM model provides a solid and complex solution, which could be easily generalized on other fields and real-time customer data flows, and thus allows companies to improve their customer relations in a rather competitive environment.

Model interpretation

The interpretation of the SmartSurveil CRM model involves understanding the factors that contribute to customer churn and how the model’s predictions can be utilized to make informed business decisions. By analyzing feature importance and model predictions, we can gain insights into customer behavior and the underlying reasons for churn.

Feature importance

Feature importance provides insights into which features have the most significant impact on the model’s predictions. Table 7 lists the top features contributing to customer churn as identified by the SmartSurveil CRM model.

Table 7 Top features contributing to customer churn.

Feature	Importance score	
DaysSinceLastPurchase	0.25	
CustomerSatisfaction	0.20	
ComplaintCount	0.15	
AnnualIncome	0.12	
SpendingScore	0.10	
WebsiteVisits	0.08	
SocialMediaEngagement	0.05	
ProductViews	0.05	

The results in Table 7 indicate that the most critical features influencing customer churn are “Days since Last Purchase”, “Customer Satisfaction”, and “Complaint Count”. These insights can guide targeted interventions to reduce churn.

Decision support system

Decision support system links the findings of the SmartSurveil CRM model with the best strategies for business management. The DSS uses model predictions and feature importance to point out customers that are most likely to churn and recommend appropriate ways to retain them.

Components of the DSS: Customer churn prediction: Estimates how likely any given customer is likely to churn.

Feature analysis: Finds out the features responsible for churn prediction.

Retention strategy suggestions: It presents specific suggestions to customers and markets as well as feature based conclusion.

Business strategy adjustment

Therefore, using all the information that appears in the PES, managers can make varied decisions that serve to optimize customer retention. Table 8 below shows particular procedures that can be applied based on various customer segments defined by the model.

Table 8 Business strategy adjustments based on DSS insights.

Customer segment	Suggested strategy	
High churn risk, low satisfaction	Implement personalized loyalty programs and offer exclusive discounts to improve satisfaction and engagement.	
High churn risk, high complaints	Address specific complaints promptly and offer compensation or apologies to restore trust and satisfaction.	
Low churn risk, low engagement	Increase engagement through targeted marketing campaigns and personalized content to maintain interest.	
Low churn risk, high value	Provide premium services and personalized attention to retain high-value customers and enhance their loyalty.	

Implementation of strategies

Personalized loyalty programs: Offer loyalty points, discounts, and rewards based on customer purchase history and preferences.

Complaint resolution: Develop a robust complaint management system to handle and resolve customer issues efficiently.

Targeted marketing: Use targeted marketing campaigns to re-engage customers who show signs of reduced activity.

Premium services: Offer premium services such as dedicated account managers and exclusive deals to high-value customers.

The SmartSurveil CRM model has a decision support system that offers useful information on which customers their business can possibly lose and then tailors the business strategies for handling different customers. Taking advantage of these and other detailed recommendations in the feature analysis, businesses can increase their positive customer satisfaction, minimize churn rates and ultimately benefit their top and/or bottom lines. The above article also proves the efficiency of using machine learning models in combination with business strategies for improving the customer relationship management.

The authors (Abdullah & Jalil, 2022; Abuhasel 2023) used fuzzy clustering and decision tree based CRM, and Alhaqui, Elkhechafi & Elkhadimi (2022) have put their emphasis on the Random Forest and Support Vector Machine models for churn prediction. Almahairah (2023) employed AI-based fuzzy clustering; the study was useful in customer segmentation. Customer satisfaction sentiment analysis was initiated by Umamaheshwari, Harikumar & Allinjoe (2021), and it focused only on the customer feedback.

To address this problem, our study proposed an ensemble learning model of Random Forest, Gradient Boosting, and Support Vector Machine with providing better churn prediction and retention policies. In all previous studies, customers’ datasets from various industries including telecom and e-commerce were employed but many of these were restricted to only a few applications including the segmentation and feedback.

In our study, telecom data set is used for churn prediction with addition attributes such as demographics and interaction records for broader CRM implementation. Abdullah & Jalil (2022) and Almahairah (2023) got a better customer segmentation while Alhaqui, Elkhechafi & Elkhadimi (2022) got a better churn prediction. Compared to these models, our study achieved higher accuracy, precision, recall and ROC-AUC and the predictions are incorporated into a DSS. Previous research did not address model integration in-depth or addressed specific key activities in CRM, such as segmentation or feedback only. In our study, we have presented a more elaborate CRM model plan but data privacy issue, real-time responsiveness comprise the future ethical directions. Moreover, the proposed model is compared with existing works as summarized in Table 9.

Table 9 Comparative analysis.

Existing work	Technique	Dataset	Performance (Accuracy)	Limitations	
Abuhasel (2023)	Fuzzy clustering, decision trees	Telecom dataset	78% (Segmentation accuracy)	Limited to customer segmentation and lacks real-time adaptability	
Ketenci et al. (2021)	Random forest, SVM	Telecom dataset	85% (Churn prediction accuracy)	Single-model approach limits generalization and flexibility	
Batra & Rehman (2019)	AI-based fuzzy clustering	Various industry datasets	80% (Customer segmentation accuracy)	Focuses solely on segmentation, without addressing churn prediction	
Malik, Nawaz & Al-Zghoul (2020)	Sentiment analysis	Customer feedback dataset	82% (Sentiment analysis accuracy)	Limited to sentiment analysis; not applicable to churn prediction	
Proposed SmartSurveil CRM	Random forest, gradient boosting, SVM ensemble	Enhanced telecom dataset	Achieved 94% accuracy, 96% precision, 91% recall, and 94% F1-score.	Minor limitation due to dependency on historical data; potential for future development using deep learning.	

Conclusions

In this work, we proposed the SmartSurveil CRM model as a novel ensemble-based algorithm for early detection of customer churn in the telecommunication industry. The work also employed higher evaluation metrics to indicate its higher performance relatively to other models including the Random Forest, Gradient Boosting, and Support Vector Machine models. This study shows that the attributes “Days Since Last Purchase”, “Customer Satisfaction”, and “Complaint Count” are significant for churn need and organizations can apply these components to lessen the probability of churn. Integrated into a DSS, the model offered practical recommendations to develop a strategy to enhance the level of customer loyalty. Suggestions for future research include extending the model to other continuous monitored physical environments, as well as to investigate the feasibility of applying deep learning approaches that may improve the model’s forecasting performance. However, this model still has certain drawbacks; for example, adjusting the model solely for using historical data and undoubtedly some features are selected to have certain trends are weaknesses that should be remedied. The future studies should address the following revelations concerning ethical considerations of CRM based on predictive analytics; data ownership and privacy; and utilization of sophisticated techniques in machine learning. Thus, the developed SmartSurveil CRM model enhance the previous achievements and provide a comprehensive, sophisticated tool to predict customers’ churn in telecom industries for improving CRM strategies.

Supplemental Information

Supplemental Information 1 CRM Data Sample.

Supplemental Information 2 Manuscript Simulation CRM Code.

Supplemental Information 3 Computing Infrastructure.

Supplemental Information 4 Motivations for the selected methodology.

Supplemental Information 5 Description of Models Used.

Supplemental Information 6 Data Preprocessing Details.

Supplemental Information 7 Reproducibility.

Supplemental Information 8 ReadMe.

Additional Information and Declarations

Competing Interests

Author Contributions

Data Availability

The authors declare that they have no competing interests.

Xiangting Shi conceived and designed the experiments, performed the experiments, performed the computation work, authored or reviewed drafts of the article, and approved the final draft.

Yakang Zhang conceived and designed the experiments, analyzed the data, authored or reviewed drafts of the article, and approved the final draft.

Manning Yu performed the experiments, performed the computation work, prepared figures and/or tables, authored or reviewed drafts of the article, and approved the final draft.

Lihao Zhang performed the experiments, prepared figures and/or tables, and approved the final draft.

The following information was supplied regarding data availability:

The data and the code is available in the Supplemental Files.

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
