# Peer review of "Revolutionizing market surveillance: customer relationship management with machine learning"

_PeerJ Computer Science, doi:10.7717/peerj-cs.2583_

## Round 0.1 · original submission · Minor Revisions

Thank you for the submission. The reviewers’ comments are now available. It is not suggested that your article be published in its current format. We do, however, advise you to revise the paper in light of the reviewers’ comments and concerns before resubmitting it. The followings should also be addressed:

1. Pay special attention to the usage of abbreviations. Spell out the full term at its first mention, indicate its abbreviation in parenthesis and use the abbreviation from then on.
2. Some paragraphs are too long to read. They should be divided into two or more.
3. Equations should be used with correct equation number. Many of the equations are part of the related sentences. Attention is needed for correct sentence formation.
4. All of the values for the parameters of all algorithms should be given.

Best wishes,

·

Basic reporting

Pass

Experimental design

No comment

Validity of the findings

No comment

Additional comments

No comment

·

Basic reporting

I believe the proposal has some merit, but I do not believe the paper in its current form demonstrates this potential as a research paper. Paper reads like an internal report instead of an academic research paper. It does not discuss alternative approaches, and it does not discuss weakness and strengths. In a scientific paper the authors have to discuss their work in context of related work and they have to elaborate what the original contribution to the state of the art is. Unfortunately, this paper fails completely in all of these aspects.


1. Introduction: Should be re-written. Currently there are some disconnected ideas that should be reorganized. The justification for the motivation of this study is relatively weak and not clearly explained. Introduction should show the research gap, paper motivation, paper purpose and which is the paper knowledge contribution to solve the research gap.

2. Literature review.
- It should be organized in subsections. In addition, paragraphs are too long. It makes the paper not easy to read. Please re-shape the paragraphs.
- “There are some specific recent researches on the topic that are worthy to consider”. Information about how these recent researches have been chosen is necessary (sources, number of citations, etc.). How can we know that they are the main contributions in the field?.
In addition, a critical analysis of the state of the art is necessary identifying the research gaps in the field.

Experimental design

Article content is within the Aims and Scope of the journal: YES
Rigorous investigation performed to a high technical & ethical standard: YES
Methods described with sufficient detail & info to replicate (code, dataset, computing infrastructure, reproduction script, etc.): YES
Is there a discussion on data preprocessing and is it sufficient/required? YES
Adequately described evaluation methods, assessment metrics, and model
selection methods: YES
Are sources adequately cited? YES

Validity of the findings

Discussion section. The contribution of the author’s approach to the literature is not highlighted. The literature review needs to be integrated with the claims that the authors make in order to show the importance of their contribution. The piece is lacking in originality or a clear contribution to the literature.
In addition, in the discussion section, authors should prove that they have achieved the research objectives.

Conclusion section: I would like that authors show better the consequences for academics and practitioners of the results. It has to be showed in the conclusion section.

·

Basic reporting

Nothing is found

Experimental design

Nothing is found

Validity of the findings

1. if possible add the settlement ratio of the complaint as a feature
2. Missing comparison between proposed work & work done.

Additional comments

It's a well-written paper that contains almost all kinds of parameters for analysis.

Reviewer 4 ·

Basic reporting

1. Abstract: the abstract needs to be reworked, the abstract should introduce the paper, put forward the aim, the methods, results, and conclusion in order.
In lines 22-24, the authors made reference to a statement which requires reference and this is not possible in an abstract. Rephrase Line 19 and rework the abstract.
2. Introduction: the authors provided a fair introduction, few of the paragraphs needs some grammar check. Specifically. Lines 63-63,
lines 74-75 “Companies cannot sustain themselves especially in the existing emerging markets fast”
Line 79- what do you mean by “Expired rules”
Line 88-89 “The failure of industries to make minor interactions with customers scalable personal; the ways of handling and analyzing large amounts of data from various sources”
Line 128-130- “The structure of this research paper is presented below, section 1 in this article is the introduction it indicates the topic of the study, the objectives, scope of the study and its importance”
Note: you already introduced the paper, I do not think you should state that here again. You should highlight only the remainder of the paper in that paragraph.
Lines 128-136 needs to be grammatically evaluated.
Lines 175-177. Please provide reference
Lines 182
The references were inconsistent. Please use the Journal referencing style and be consistent in your referencing pattern.

I do not see a reason for figure 1.

Experimental design

1. How was the data collected, you stated that the data were telecom data. It is advisable to include the source of the data.
2. Before preprocessing, was there any form of filtering?
3. What is the size of the dataset?
4. What percentage was allocated to training and testing?

Validity of the findings

1. The authors adopted random forest for the proposed model, what is the justification for using RF? You need to justify your choice of selection.
2. Please refer to the tables properly. For example, say, Table 3 explains… instead of saying the following table.
3. The articles sheds new light on the discourse.
4. The evaluations are well performed, excepts for the questions raised

Additional comments

1. Discussion and results please align you r results with the literature and showcase the uniqueness of your study in the discussion and conclusion section.
2. The paper needs to be grammatically edited.
3. Lines 248-250 are duplicated.
4 A proof read and English editing is required

---

## Round 0.2 · Minor Revisions

Dear Authors,

Thank you for the submission. The reviewers’ comments are now available. It is still not suggested that your article be published in its current format. We do, however, advise you to revise the paper in light of the reviewers’ concerns and comments.

Best wishes,

·

Basic reporting

As I said in my previous review, I think the proposal has some merit. However, although authors have made a good revision, I still have some concerns that they have to overcome.

1. Abstract. I think that abstract is too long. Authors should try to synthetize
It
2. Proof read and English editing. Authors say that they have done it. However, I still think that the writing style can be better. There are some long paragraphs (e.g. 4.4. discussion) and some typos.
In addition, I think that introduction section can be improved. I still don’t like the flow of this section. For example, in line 125 authors say “SmartSurveil CRM model proposed here…” but in line 134 authors say “Current practices in market surveillance and customer relationship management do not suffice…”. The flow should be: the research gap, paper purpose and which is the paper knowledge contribution to solve the research gap. Now, you are showing paper contribution before the diagnosis of the stat of the art

Experimental design

Discussion. Discussion shouldn’t be inside the results section. It should be a separate section. In addition, authors cannot say “We will further expand this section in future versions of the paper to provide a clearer contrast with prior research.” They have to show it in this version of the paper

Validity of the findings

They are OK

Reviewer 4 ·

Basic reporting

The paper has evolved significantly and I am comfortable with the response from the authors. However, the authors claimed that the data was sourced from a telecom internal database, which brings me to the following question
1. what are the ethical concerns in your data-gathering process and how were they addressed?

Experimental design

N/A

Validity of the findings

N/A

Additional comments

The paper has evolved significantly and I am comfortable with the response from the authors. However, the authors claimed that the data was sourced from a telecom internal database, which brings me to the following question
1. what are the ethical concerns in your data-gathering process and how were they addressed?

---

## Round 0.3 · accepted · Accept

Dear Authors,

Thank you for revising the paper. The reviewers thik that your paper can be accepted in this final form. The paper has been sufficiently revised and is now ready for publication.

Best wishes,

·

Basic reporting

The authors have modified some items of the paper according to the reviewers’ comments. For this reason, I think this last version of the manuscript is ready to be published.

Experimental design

The authors have modified some items of the paper according to the reviewers’ comments. For this reason, I think this last version of the manuscript is ready to be published.

Validity of the findings

The authors have modified some items of the paper according to the reviewers’ comments. For this reason, I think this last version of the manuscript is ready to be published.